# Cognitive Impairment and Dementia Data Model: Quality Evaluation and Improvements

**Dessislava Petrova-Antonova** * and **Sophia Lazarova**

GATE Institute, Faculty of Mathematics and Informatics, Sofia University "St. Kliment Ohridski",
1504 Sofia, Bulgaria
* Correspondence: d.petrova@fmi.uni-sofia.bg

**Abstract:** Recently, datasets with various factors and indicators of cognitive diseases have been available for clinical research. Although the transformation of information to a particular data model is straightforward, many challenges arise if data from different repositories have to be integrated. Since each data source keeps entities with different names and relationships at different levels of granularity and format, the information can be partially lost or not properly presented. It is therefore important to have a common data model that provides a unified description of different factors and indicators related to cognitive diseases. Thus, in our previous work, we proposed a hierarchical cognitive impairment and dementia data model that keeps the semantics of the data in a human-readable format and accelerates the interoperability of clinical datasets. It defines data entities, their attributes and relationships related to diagnosis and treatment. This paper extends our previous work by evaluating and improving the data model by adapting the methodology proposed by D. Moody and G. Shanks. The completeness, simplicity, correctness and integrity of the data model are assessed and based on the results a new, improved version of the model is generated. The understandability of the improved model is evaluated using an online questionnaire. Simplicity and integrity are also considered as well as the factors that may influence the flexibility of the data model.

**Keywords:** data modeling; data model quality evaluation; cognitive impairment; dementia; interoperability of clinical data

## 1. Introduction

Cognitive disorders, especially dementia—a condition severe enough to compromise social and/or occupational functioning [1]—have a huge social significance to all parties involved in their diagnosis, treatment and caretaking. Patients undergo an ever-progressing cognitive decline and gradually lose independence, which puts a heavy burden on their caregivers, including family members, hired professionals or the personnel of specialized institutions. As the number of cases increases with every decade, cognitive disorders present a huge public and financial burden. Finally, they also pose several still unanswered moral and ethical issues. Like most mental disorders, cognitive disorders are caused by various factors and brain problems such as single or repeated head injuries, infections, toxicity, substance abuse, benign and malign brain tumors, many genetic diseases, etc. Most of them are caused by neurodegenerative diseases, vascular damage to the brain and the various combinations between them.

According to the World Health Organization (WHO), around 55 million people worldwide have dementia, with over 60% living in low- and middle-income countries [2]. This number is expected to rise to 78 million in 2030 and 139 million in 2050 [2]. The increased dementia morbidity is due to the tendency for increased life expectancy since each decade of human life exponentially increases the chance of developing dementia [3]. In fact, dementia has been recognized as a public health priority by the WHO, which only underlines the need for centralized strategies to combat the devastating effects of the disease. As a

result, several global plans were developed in order to improve areas such as dementia awareness, reducing dementia risk, early diagnosis, effective treatment and care, as well as research and innovation [4–6].

A central prerequisite for the active research of cognitive diseases is data availability. Leveraging the collection and storage of large data volumes could open new horizons in this area of research since it is the basis of data analyses and data insight discovery. The digitalization of medical data, particularly in cognitive diseases, has the opportunity to not only ease clinical management of those diseases by creating registries of patients, enabling strict follow-up and creating a robust schedule of cognitive rehabilitation but also to hugely amplify the ability to conduct large-scale research by applying Big Data and Artificial Intelligence (AI) technologies. Many digital repositories have been created worldwide to support research on cognitive diseases [7]. They store a variety of factors and indicators of the patient history and current status. Although the transformation of information to a particular data model is straightforward, many challenges arise when data coming from different sources and formats has to be integrated. Since each data source keeps entities with different names and relationships at different levels of granularity, a part of the information can be lost or not properly presented. To avoid incorrect data transformations and enable data integration from various sources, we propose a common hierarchical data model, featuring the following benefits for researchers and clinicians [8]:

- Provides semantics of the data in a human-readable format and accelerates the interoperability of clinical datasets;
- Suitable for use as a stand-alone data model for clinical data as well as a middleware for mapping between different data models;
- Provides a foundation for implementing data schemas across different types of databases and further system development;
- Enables the application of Machine Learning (ML) and AI algorithms and models by helping data scientists to understand the data and select appropriate features.

Even though the data modeling phase occupies a small proportion of the total development effort, its influence on the resulting system is probably greater than any other phase [9]. Since conceptual data modeling is the first step of database design, the quality of the data model is a major determinant for the quality of the database and the overall quality of the respective informational system [10]. Therefore, building quality data models is central to building quality systems.

This paper extends our previous work [8] by presenting a refined data model that is a result of a quality evaluation process based on a set of quality factors and metrics. The improved version of the data model is obtained by assessing the completeness, simplicity, correctness and integrity of the initial version of the data model. In addition, the understandability of the improved model is evaluated using an online questionnaire. Simplicity and integrity are also considered. Finally, the factors that may influence the flexibility of the data model are analyzed.

The rest of the paper is organized as follows. Section 2 provides a brief overview of the original version of the data model. Section 3 presents the methodology followed for the quality evaluation of the data model, while Section 4 outlines the outcomes of this evaluation. Section 5 describes the obtained results. Section 6 gives conclusions and directions for future work.

## 2. Background

The data model for cognitive disease assessed in the present work was originally presented and described in [8]. The model was built following the existing domain literature and in close collaboration with clinicians. In all, 57 requirements were collected and further applied during its elaboration; they are listed in Appendix A (see Table A1). The formalization of the data model was completed using Unified Modeling Language (UML) [11], YAML Ain't Markup Language [12] notations and corresponding software tools.

The resulting data model defines entities, attributes and relationships needed to create a patient's profile. It is developed to unify and structure the information relevant to four main domains of the profile—personal data, medical history data, objective clinical investigations and treatment prescribed (Figure 1).

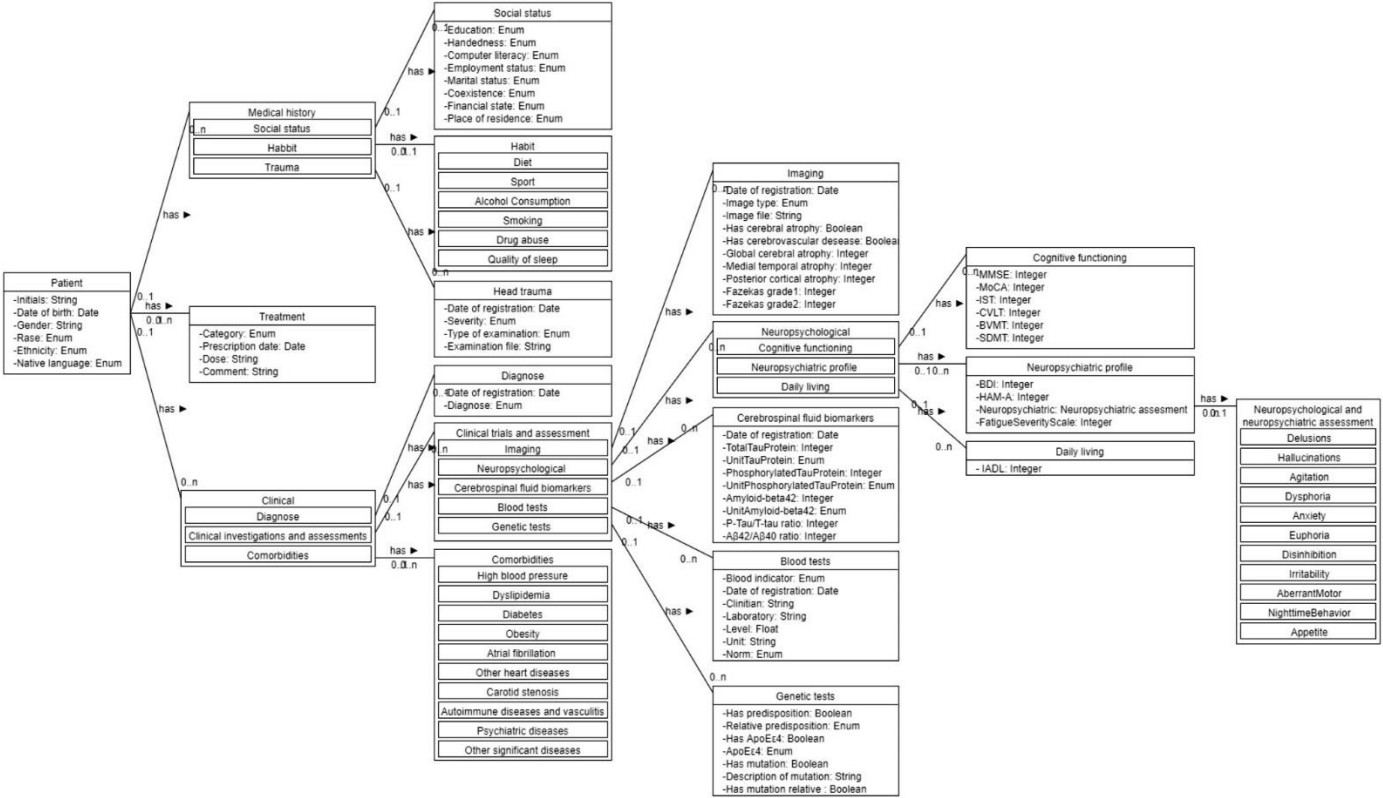

**Figure 1.** UML diagram of the data model for cognitive diseases.

The Personal profile includes the patient's personal data. The personal data is modelled as a Patient entity, including attributes such as date of birth, gender, race, ethnicity, and native language. The medical history data presented in the Anamnestic profile is related to the patient's social status, everyday habits, and head traumas, modelled as separate entities. The Clinical profile is described with data about medical investigations and assessments, comorbidities and their severity, and ultimately, the most likely diagnosis. It covers six aspects of diagnostics: imaging, neuropsychological and neuropsychiatric assessment, cerebrospinal fluid biomarkers, blood tests, genetic data, and comorbidities. The Treatment profile is related to medications prescribed to the patient. The medications are divided into groups: medications for degenerative cognitive disorders; medications for cerebrovascular disease; antiplatelet and anticoagulant drugs; neuroleptic drugs; antidepressant drugs; and medications for sleeping. Each medication is described with the prescription date and the daily dosage in milligrams.

## 3. Materials and Methods

The simplest frameworks for evaluation of data model quality are based on defining quality as a list of desirable properties [13]. However, these lists are often unstructured and burdened by loose or imprecise definitions [14]. A more comprehensive approach to quality evaluation are theoretical frameworks which define central concepts that underlie the quality of models. For example, Lindland et al. proposed such framework based on semiotic theory [14]. For each semiotic level (syntactic, semantic, pragmatic) this framework offers not only quality goals but also means to achieve them. There is also an extended version of this framework including a fourth semiotic level [15]. However, these

frameworks appear to be rather general since they are dealing with conceptual models in general, not just data models. Furthermore, since the data model evaluated here seeks to address existing practical needs of the medical domain, we were interested in quality evaluation procedures that had been previously applied in real-world projects.

Therefore, to assess the quality of the data model for cognitive impairment and dementia we followed the quality factors and metrics outlined in the framework for quality evaluation of data models proposed by Moody and Shanks [9,16]. While the majority of the methodologies addressing data model quality have emerged from theoretical work, this framework was developed as a result of years-long practical experience in quality assuring data models [16]. Furthermore, the framework was empirically validated through a combination of field and laboratory research methods including application in two of the largest commercial organizations in Australia [16,17].

The framework defines a set of quality factors that can be used to evaluate the quality of individual data models, namely—completeness, flexibility, understandability, simplicity, integration and implementability. Importantly, these factors were assembled to incorporate the needs of all stakeholders and to capture the various aspects of data model quality. In later work, the framework was used to develop advanced metrics for entity model quality evaluation and two additional quality factors were added—correctness and integrity [18].

We did not use the complete set of quality factors in our evaluation process. Particularly, there were two factors that we did not include—integration and implementability. Integration is defined as the level of consistency of the data model with the rest of the organization's data [18]. However, our model was not created for the needs of a particular organization and therefore such evaluation is not applicable at this point of time. The case for implementability is similar. Implementability describes the ease with which the data model can be implemented within given project parameters such as time limits, budget and technological constraints [18]. Since our data model is currently not set to be implemented by a particular organization, we cannot perform such an evaluation.

### 3.1. Procedure

First, we evaluated the completeness, simplicity, correctness and integrity of the original data model shown in Figure 1. Then, we used the results to generate an improved version of the same data model. Finally, we evaluated the understandability of the improved model with the means of an online questionnaire. Additionally, we also evaluated the simplicity and the integrity of the improved model. At the end, we offer analysis of the factors that may influence the flexibility of the model. Since UML diagrams are a well-known and commonly used medium for communication of data models, we chose to evaluate our data models based on their UML representations.

The next paragraphs describe the quality factors we employed in our assessment as well as the metrics that were used for the evaluation of each quality factor.

### 3.2. Completeness

Completeness refers to the extent to which the data model represents all user requirements accurately. Therefore, completeness can be assessed by demonstrating that each requirement is represented somewhere in the model while each element of the model corresponds to a user requirement [18]. Thus, we are evaluating the completeness of the model by identifying the mismatches between the user requirements and the data model. According to Moody there are several types of mismatches between user requirements and data models [18]:

- Type 1 error—items that do not correspond to user requirements;
- Type 2 error—user requirements that are not represented in the data model;
- Type 3 error—items that correspond to user requirements but are inaccurately defined.

The number of occurrences of these errors was used as metric of completeness. The lower the number of mismatches, the higher the level of completeness.

### 3.3. Integrity

Integrity is defined as the extent to which the business rules or integrity constraints which apply to the data are enforced by the data model [18]. In the context of our model, such rules are value constraints of attributes and multiplicity requirements. However, such rules are already part of the requirements accompanying the data model and thus the evaluation of integrity is essentially part of the completeness evaluation. Nevertheless, we treat integrity as a separate quality that is coupled with completeness.

### 3.4. Correctness

Correctness is concerned only with whether the data model is a result of accurately used techniques for data modeling. Among others, rules of correctness include diagramming conventions, naming rules, definition rules and rules of composition [18]. Additionally, correctness is ensuring that the model contains no redundancy.

Since correctness is rooted in evaluating the proper application of UML diagramming conventions and practices, we defined three possible groups of violations—major violations, minor violations and redundancies. The number of occurrences of these errors was used as a metric of correctness. Major violations are any violations of diagramming convention that directly compromise the representational quality of the model. Examples are inaccurate usage of relationships, inaccurately set multiplicities, etc. We define minor violations as violations that do not impact the representational quality of the model. For example, violations of naming conventions. Finally, as redundancies we define items that are present in the diagram but are unnecessary or defined in an overly complicated way.

### 3.5. Simplicity

Simplicity refers to having a minimal number of constructs within a model. In the majority of the cases the simplest model appears to be the best model [9]. In general, simpler models are not only easier to implement and to understand but also more flexible [9]. Therefore, simplicity is a very desirable quality that is often thought to be central to satisfying many other data modeling quality requirements. Even though it is quite easy to evaluate this metric, there are multiple methods that can be used for its evaluation [18].

We used the complexity of the model, measured by the number of entities plus the number of relationships within the model (E + R), as an approximation of simplicity. This metric is based on complexity theory which postulates that the complexity of a system can be measured by the number of components plus the number of relationships between the components [9]. Since we are searching for the simplest model possible, we are looking to minimize the E + R metric.

### 3.6. Flexibility

Flexibility describes the ability of the model to cope with future changes in the business requirements. High level of flexibility ensures that future changes in requirements can be handled with minimum modifications to the data model. This results in low maintenance costs and higher organizational responsiveness [18].

Flexibility is a quality that is particularly hard to assess since it is closely related to changes taking place in the future. However, one can still evaluate the potential likelihood of future business changes that might lead to modifications of the data model. Therefore, flexibility is estimated via the number of elements in the model that might be subject to change in the future. We defined three types of possible future modifications that might occur in the model—deletions, additions and alterations. Then, we analyzed the plausible changes in the domain and how they would affect the model in terms of the outlined types of modifications. Since the results of the flexibility evaluation are expected to be nearly identical for the initial version of the model and the improved version, we are offering in-depth flexibility analysis only for the latter one.

### 3.7. Understandability

Understandability is one of the most commonly used metrics in the quality evaluation of data models [19]. It is critical for a data model to be well understood by business users and application developers. Alternatively, poorly understood data model is likely to result in a poorly understood system or a system that does not meet business and user requirements.

We evaluated the understandability of the improved model by assessing the ability of a user group to interpret the model correctly. For this purpose, we used two measures of actual understanding—problem-solving task and a cloze test.

The participants were instructed to complete the cloze test and the problem-solving task by using the provided UML diagram of the data model for reference. Importantly, the UML diagram was at the disposal of the participants until they were finished with the questionnaire. The evaluation was conducted via an online questionnaire implemented with Google Forms. The questionnaire included three sections—participant profile and baseline knowledge assessment, cloze test component and problem-solving component. There were no time limitations.

The Cloze test (fill-in-the-blanks task) consisted of a short text describing the data model that had 14 missing words in it. The participants were asked to fill in the blanks with the appropriate words or type in 'IDK' in case they could not think of any word that would fit. The Cloze test was given first because the passage used in the task had a descriptive nature and thus it provided an overview of the model and the UML diagram. The complete text and the list with the accepted answers are available in Appendix A. Note that for some of the blanks synonyms and words that bear similar meaning were also accepted.

The problem-solving part consisted of situational questions that required interpretation of the UML diagram and reasoning with its content. The last two questions of this section were constructional—they asked the participants to modify the UML diagram in order to implement additional requirements. While one of the constructional questions required only modification of the already created classes the other one required both modification of existing classes and addition of new ones. Each question from this section required a motivated, open answer. Each answer was scored on a scale between zero and two where zero indicated an incorrect answer, one indicated a partially correct answer and two indicated a fully correct answer. Grading was conducted through the formulation of base answers. A base answer is such an answer that is exhaustive and minimal. We formulated a base answer for each question and then compared the answers given by the participants to the base answers. If the gist of a given answer and the base answer matched completely, the answer was graded as correct. If the match was only partial, the answer was graded as partially correct. The questions with their respective base answers are listed in Table A3, Appendix A.

In all, 62 individuals took part in the online evaluation. They are students at the Faculty of Mathematics and Informatics at the Sofia University, pursuing studies in the field of computer science and software engineering. All of them attended to courses related to semantic technologies, data structuring and modeling, database design and implementation, and have background knowledge in object-oriented modeling and data modeling patterns.

## 4. Evaluation of the Original Model

This section presents the evaluation of the initial data model according to the methodology described in Section 3.

### 4.1. Completeness

To evaluate completeness, we identified the number of mismatches between the data model and the list of requirements that was created a priori. There were 57 requirements and each of them was evaluated separately.

The evaluation of completeness resulted in identifying seven errors of Type II, 13 errors of Type III and two errors of Type I. The evaluation also showed that 65% of the requirements were implemented correctly (Table 1). The full list of requirements along with the committed errors are available in Table A1.

**Table 1.** Results from the completeness evaluation. The table shows the number of errors identified during the evaluation, the number of correctly implemented requirements and the total number of requirements. Type I error—items represented in the model but not corresponding to the requirements; Type II error—items represented the requirements but not in the model; Type III error—items that are represented in the model and correspond to the requirements but are defined inaccurately in the model.

| Type I Errors | Type II Errors | Type III Errors | Correctly Implemented | Total Requirements |
|---|---|---|---|---|
| 2 | 7 | 13 | 37 | 57 |

The results suggest that the model has average level of completeness that can be significantly improved by addressing the identified errors. The relatively low number of Type II errors shows that the majority of the requirements were represented in the model in some way. However, the significant presence of Type II errors suggests that either the requirements were defined in an ambiguous way or the communication with the domain experts was insufficient or unclear. In fact, it has been previously established that individuals with different backgrounds tend to use different communication styles and vocabularies, which often causes difficulties during collaboration [20]. Therefore, our results are emphasizing the need for clear communication and effective knowledge sharing practices in interdisciplinary environments.

*4.2. Simplicity*

The model is characterized by a large number of entities and a significantly lower number of relationships (45 entities and 18 relationships). Such a prominent imbalance suggests the presence of architectural deficiencies in the diagram such as incomplete representations of relationships and entities. In fact, Figure 1 shows 17 nested classes that are not defined as separate entities connected with relationships to their owner class. Instead, they are only listed within the owner class (Figure 1, *Comorbidities* class). This manner of partial representation is leaving a large number of relationships hidden. Taking all classes with impartial definitions and defining them properly would add another 17 relationships to the diagram. Thus, the actual simplicity of the diagram would be much lower than what the eye meets since the actual value of the E + R metric is not 63 (E = 45, R = 18) but 90 (E = 45, R = 45). In turn, this shows a high level of ambiguity due to improper use of UML diagramming rules.

*4.3. Correctness*

To estimate the correctness of the model we analyzed the architectural and structural quality of the UML diagram (Figure 1). We identified five major violations, two minor violations and five redundancies (Table 2). While some of the violations were isolated to one or several occurrences, there were systematic violations that occurred consistently across the diagram. For example, despite having multiple attributes of type enumeration in the diagram, none of the enumerations is actually defined. Another systematic violation is the overall lack of attribute constraints.

**Table 2.** Violations identified during the evaluation of correctness. The identified violations are separated in three groups—major violations, minor violations and redundancies.

| Major Violations | Minor Violations | Redundancies |
|---|---|---|
| **Using association where composition is needed**. Example: *Treatment* is part of *Patient* and should exist as long as *Patient* exists. Currently this relationship is not represented in the diagram. | **Violation of naming convention I** Incorrect naming of classes (upper camel case) and attributes (lower camel case); | Genetic testing is essentially a blood test. It does not have to be a separate entity. |
| **Incorrect multiplicities** Example: Every *Patient* should have a single *Medical History* record. Currently a *Patient* can have any number of *Medical History* instances. | **Violation of naming convention II** Classes cannot be named with adjectives: example—class *Clinical*. | Entity 'Clinical' fails to act as a proper superclass and therefore it is redundant. |
| **Enumerations not properly defined** Enumerations should be defined as separate entities with their possible values described. Whenever an entity has an attribute of type Enum, there should be also a relation between the corresponding enumeration and entity. | **Typos** Example: Habbit instead of Habit; Rase instead of Race | NPI items (11 items) defined as entities when they would be better defined as attributes. |
| **Missing attribute constraints** Attributes should have constraints describing the possible ranges of the attribute values. | | Types of comorbidities (10 comorbidities) defined as entities when they would be better defined as attributes. |
| **Incorrect representation of nested entities. Missing definitions for some nested entities**. All entities nested in *Comorbidities* and *Habit*. Nested entities should also be depicted as fully expanded entities that are related with the entity owner. | | *Habit* entity has six nested classes that would be better defined as attributes. |

The presence of five major violations suggests that the representational quality of the model was somewhat compromised as the model clearly failed to properly communicate some of the user requirements due to inaccurately used or defined UML components. This is in line with the results from the completeness evaluation that showed a high number of Type III errors. It is possible that the majority of these errors are due to applying UML diagramming practices improperly.

Additionally, we found five redundancies affecting one or multiple entities. A prominent example of this is the definition of *Comorbidities*—the presence of each comorbidity is encoded as a separate entity when this can be better represented with attributes. Instead of having class *Comorbidities* with 10 nested classes we could have a class *ComorbidityProfile* with 10 attributes, each denoting the presence or absence of a specific comorbidity. Similarly for *Habit* and *Neuropsychologycal and Neuropsychiatric Assessment*. The presence of so many instances of redundancy support the conclusions from the simplicity evaluation and once more points to the need for revision of the number of entities. Addressing such redundancies would improve the simplicity of the model and reduce ambiguity.

*4.4. Integrity*

Integrity deals with the correct representation of business rules in the data model, namely with the constraints on certain values. However, the correctness evaluation found that the initial UML diagram is missing attribute constraints altogether. Thus, the integrity of the model cannot be assessed accurately. In terms of integrity evaluation, we can conclude that the integrity of the model is low. In turn, this is also likely to affect the completeness of the model since these value restrictions were described in the requirements.

**5. Results and Discussion**

After analyzing the discovered deficiencies, we updated the data model to address all of the identified imperfections. This way, we obtained an improved version of the data model (Figure 2).

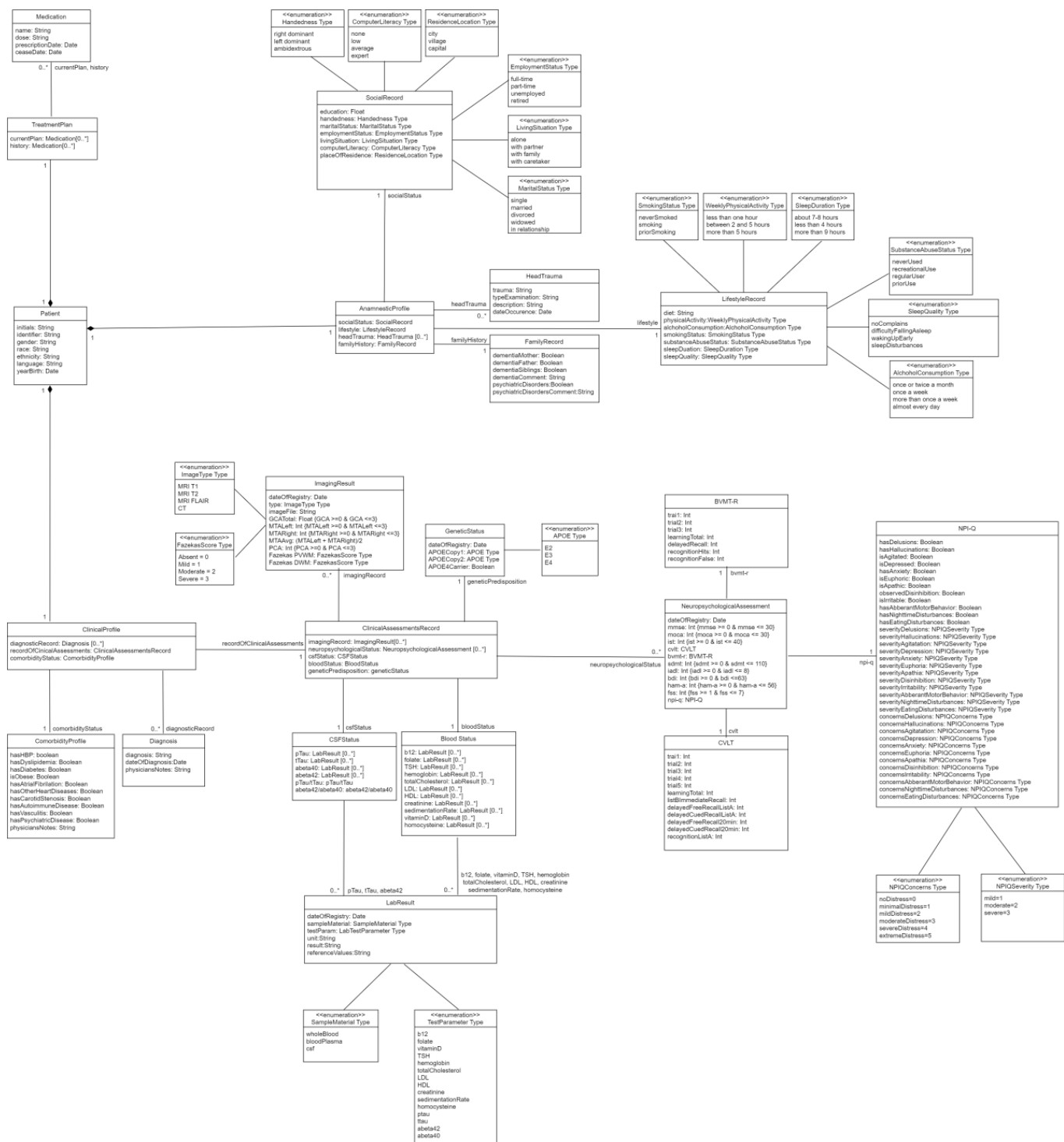

**Figure 2.** UML diagram of the improved data model.

The improved version of the model corrects all deficiencies identified in the original model in terms of completeness and correctness. Therefore, the improved model inherently scores high on both completeness and correctness. The high level of completeness guarantees that the new version of the model has a high correspondence with the initially set requirements and therefore the specifics of the domain are correctly represented within the model. The original model was found to suffer from a high number of Type III errors, the majority of which were found to be failures to apply value constraints to attributes or failures to define values of enumerations (Table A1). In most of the cases the required

attributes were present in the diagram, but their value ranges or value types had improper definitions (Figure 1, Table A1). These deficiencies had direct impact on the integrity of the model since adhering to the business requirements is what integrity is concerned with. Therefore, our results show that by maximizing the completeness of the model we are guaranteed to have a high level of integrity. This is in line with previous work on relational databases where integrity was expressed as a function of completeness and validity [21], as well as the original definition of the Moody and Shanks framework where integrity was not treated as a separate quality factor [9]. Interestingly, the completeness evaluation found two elements present in the diagram that did not correspond to any of the requirements (Type I errors). Namely, 'listing the financial state of patients' and 'listing the name of the physician who ordered a blood test' (Figure 1, Table A1). Both of these listings might be considered as a disclosure of sensitive information. Furthermore, disclosing names of attending physicians might lead to compromising the anonymity of the patients and thus cause an implicitly violation of requirement 13 (Table A1). This result emphasizes on the importance of preventing such implicit violation of the requirements from happening. While such dependencies might be subtle and sometimes easy to miss, performing a simple evaluation of completeness can significantly reduce their occurrence.

By ensuring a high level of correctness the improved model features minimized levels of ambiguity thus being more manageable and easier to comprehend. The original model was characterized by a pronounced presence of major violations and redundancies. Ultimately, this was the main source of complexity and ambiguity within the model. The failure to properly define nested classes and connect them to the hierarchy produced a hidden level of complexity thus making the UML diagram appear simpler than it actually was (Table 2). This is a prominent example of how misapplied UML practices can lead to misleading visual representations; an unfortunate phenomenon that can be prevented by strict following of the UML conventions and mandatory evaluations of correctness. The high number of redundancies, on the other hand, was the main source of complexity within the model. As a result, the original model was not a minimal model, instead it contained more than 20 redundant entities, which ultimately compromised its simplicity.

Finally, the produced results highlight the existence of a relationship between the measured quality factors. In particular, we saw that by maximizing the level of completeness we automatically improve the integrity of the data model. Additionally, ensuring a high level of correctness led to a high level of simplicity. Therefore, our results suggest that some of the quality factors might be precursors to others thus maximizing them would result in overall improvement of the model.

### 5.1. Simplicity and Integrity

The improved version of the model has a reduced number of classes compared to the original UML diagram—45 classes in the original diagram compared to 21 classes in the updated version (Table 3). This reduction is a result of addressing the redundancy issues highlighted during the correctness evaluation of the model (Table 2). Overall, the updated model has a significantly lower complexity with the E + R metric being reduced from 90 to 42 in the new version (Table 3).

**Table 3.** Simplicity metrics for the initial version of the model and the improved version. The initial model has 17 nested classes missing explicit relationships to their owner class. As a result, the explicit number of relationships in the model is 18 but the actual number of relationships is 45. Similarly for the entities + relationships (E + R) metric.

| Model Version | Number of Entities | Number of Relationships | E + R |
|---|---|---|---|
| Original model | 45 | 18 (45 actual) | 63 (90 actual) |
| Improved model | 21 | 21 | 42 |

Since the original version of the model was lacking attribute constraints, analyzing the integrity of the model was virtually impossible. As a result, the level of integrity of the initial model was virtually zero. This said, simply having a high level of correctness guaranteed in the improved model already brings somewhat of an improved integrity. Since business rules are essentially requirements and the improved model inherently features maximal completeness, we can conclude that the integrity of the model is naturally following from the high level of completeness.

### 5.2. Understandability

Out of a total of 62 participants, two were excluded from the analyses due to lack of previous knowledge in UML diagramming. Another 14 participants were excluded due to answering in inconsistent or unrelated ways, indicating that they did not spend time answering the questionnaire. Instead, it is likely that they have gone through the questions typing anything so they could submit the form faster. This led to an analytical sample of 46 participants. In all, 41 (89.1%) of them were bachelor students in their third year of studies, four (8.7%) were master students and one (2.7%) had previously completed a Ph.D.

In terms of UML knowledge, 6.5% of the participants indicated having very little knowledge of UML and 84.8% stated that they were somewhat familiar and had some practical experience with UML diagramming. Another 8.7% of the participants reported being confident in their knowledge of UML diagramming. More than half of the participants reported having no or a very little knowledge of common medical concepts such as anamnesis, medical history, brain imaging. The rest reported common level knowledge or above with none reporting high level of familiarity. Similarly, more than half of the participants reported having no or little knowledge about cognitive diseases with the rest reporting average or above average knowledge. Overall, about 15% of the participants reported having no knowledge of common medical concepts and about 22% reported no knowledge of cognitive diseases.

The mean score on the Cloze test was above average (M = 8.87; SD = 2.93) suggesting an acceptable level of comprehension in terms of the model's structure. While the Cloze test was testing the structural comprehension of the diagram, the problem-solving task required reasoning with the content of the diagram or hypothetical augmentation of the diagram to fit a requirement. Thus, the results from the problem-solving task can be used as an approximation of diagram understandability but also as approximation of domain understandability based on the interaction with the model. Each question in this section required an open answer and therefore each answer was evaluated and graded as correct, partially correct, incorrect or failed to answer (Figure 3). On average, 16.30% of participants per question failed to give an answer and 17.53% gave an incorrect answer. However, there is a high variability between questions in terms of incorrect answers and failed to answer participants. For example, only 4.35% participants failed to give an answer to Q2 and about 27% failed to provide an answer to Q4. An even more pronounced difference is observed between the number of incorrect answers given to Q3 (43.48%) and Q5 (5.22%). The high variability suggests that the participants had clear difficulties with some of the questions while others were perceived as fairly simple. While a high number of 'failed to answer' participants suggests either inability to comprehend the question or inability to interpret the diagram, a high number of incorrect answers suggests incorrect comprehension of the question or incorrect interpretation of the diagram. However, caution should be applied in the interpretation of such results since they are firmly coupled with the domain knowledge of the participant. This consideration is in line with the findings of a study that explored the common mistakes that students commit in UML diagramming. In particular, this study found that over 65% of all mistakes present in the UML models were attributed to insufficient understanding of the respective domain [22].

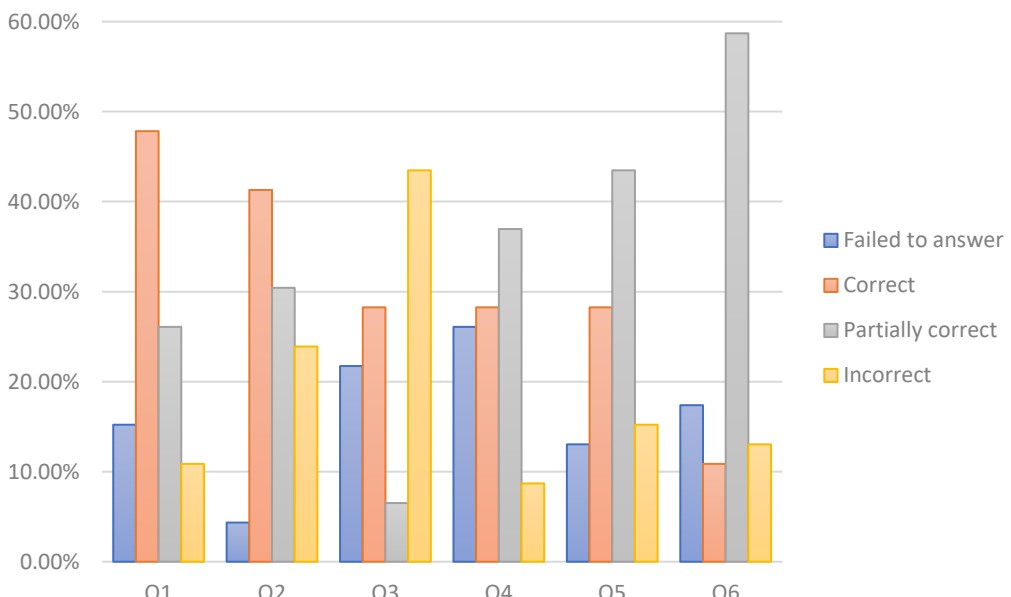

**Figure 3.** Results from the problem solving task. The table presents the distribution of failed to answer participants and participants who gave correct, partially correct or incorrect answer.

For example, on Q3 the participants had an obvious worsening of the performance compared to the other questions −21.74% failed to answer and another 43.48% gave incorrect answers. While Q3 was not designed to be of higher difficulty, it is possible that the presence of strictly medical terms such as 'PET imaging', 'positron emission tomography' and 'brain metabolism' caused confusion among the participants (Table A3). In fact, Q3 is the only question that contains specialized medical terminology related to imaging, which might explain the lower scores on this question.

On average 64% of the participants answered correctly or partially correctly to each question, with Q3 being the only question scoring below the average (34.78%). The high proportion of correct and incorrect answers suggests that the participants were not only able to understand the content of the UML diagram but also to successfully reason with it. While Q1 and Q2 received predominantly correct answers, Q4, Q5 and Q6 received an overwhelming number of partially correct answers (Figure 3). Since Q5 and Q6 were constructional questions it appear that even though the participants were able to demonstrate acceptable orientation in the diagram and the domain, they still had difficulties with properly extending the diagram to fit a new requirement. This phenomenon might be a consequence of the variability in the UML knowledge between the participants and the fact that the majority of the participants did not describe themselves as confident in their UML knowledge. Furthermore, the mass presence of partially correct answers to the constructional questions (Q5 and Q6) is in line with previous research with students and their experience with UML diagramming. In particular, a study from 2011 found that one of the greatest difficulties that students had while creating UML diagrams was the identification of the appropriate classes and relationships within the problem domain. As a result, the study reported that only 6% of the students were able to produce class diagrams with all of the expected classes and 58% of the produced diagrams had missing attributes [23].

The results from the evaluation of understandability suggest that the participants were generally able to interpret the UML diagram correctly therefore implying that the understandability of the data model is at the very least acceptable. Importantly, since the majority of the participants were students, they cannot be considered as experts in data modeling and UML diagraming. Nevertheless, the obtained results are encouraging since an expert group is expected to perform better than our current sample.

### 5.3. Flexibility

The analysis of flexibility showed that the data model for cognitive impairment and dementia is characterized by a high level of flexibility which is a product of two main factors. The first factor has to do with the characteristics of neurology, namely the fact that while neurology is constantly advancing on academic and research level, changes are being slowly adopted in clinical practice. Therefore, despite the active clinical research in the field, usually it takes a solid amount of time for innovations in methods, tests and treatments to be implemented in the clinical practice. Second, our data model was designed as a general model that can be used for managing patients with cognitive disorders. Thus, it possesses an inherent level of abstraction and flexibility within the domain of cognitive disorders.

Considering the fact that clinical guidelines are listing only proven to be beneficial practices, there is a very low likelihood of having any of the already established assessments removed. Therefore, we do not anticipate any significant deletions to occur within the model, even if there is an update in any of the guidelines regarding cognitive disorders.

On the other hand, there is ongoing research aiming to find cheaper and less invasive biomarkers for various diseases, including neurodegenerative diseases. For example, more and more studies are putting forward the idea of blood-based biomarkers for Alzheimer's disease. Even though blood-based biomarkers are not available for clinical use, there is an accumulating body of literature demonstrating their value as complementary biomarkers [24–27]. Thus, blood-based biomarkers are expected to soon become available to clinicians as a cheaper and less invasive alternative to CSF biomarkers and PET imaging [28]. Considering such examples, we acknowledge the high possibility of a future need for additions to our model. However, the extent of such additions will be limited to several entities and thus they will require low amounts of effort and minimal modifications.

The final kind of possible modifications to the data model that we discussed are alterations. Alterations consist of small changes that do not interfere with the number of entities or relationships in the model. A necessity of such modifications may occur in the future, especially after the model is implemented in a system and deployed for user testing. Nevertheless, alterations would require minimal changes in terms of time and effort.

### 6. Conclusions and Future Directions

The present work presented the results from a quality evaluation of a data model for cognitive impairment and dementia. By addressing the issues found during the quality assessment we produced an improved version of the model that has inherently high levels of completeness, correctness, integrity and simplicity. The understandability and the flexibility of the improved model were also considered.

While the improved model was found to be of high flexibility, we were not able to conclusively determine the understandability of the model. Nevertheless, what we were able to infer from the results was that the understandability of the model is, at the very least, acceptable. This limitation is due to the characteristics of the sample used for the evaluation of understandability—university students. Thus, future work might focus on a more elaborate evaluation of understandability, performed by experts in data modeling and UML diagramming.

Furthermore, our results highlight a possible connection between some of the quality factors. In particular, completeness and correctness serving as determinants of integrity and simplicity, respectively. While the results appear to suggest that the number of quality factors used for evaluation may be reduced without reducing the validity of the approach, further work is needed to establish and confirm such connection.

Finally, future work includes the development of ontology of cognitive diseases as an expansion to the data model, followed by the implementation of a graph database based on the ontology. Ultimately, the database is going to integrate data from different sources providing the basis for advanced data analytics. Furthermore, a dedicated Extract Transform Load (ETL) procedure and a corresponding tool are planned to be developed in

order to import data from large dementia studies. As a result, we will be able to populate the data model and validate it with actual medical records.

**Author Contributions:** Conceptualization, D.P.-A. and S.L.; methodology, S.L.; validation, D.P.-A. and S.L.; formal analysis, S.L.; investigation, D.P.-A. and S.L.; writing—original draft preparation, S.L.; writing—review and editing, D.P.-A.; visualization, D.P.-A. and S.L.; supervision, D.P.-A.; project administration, D.P.-A.; funding acquisition, D.P.-A. All authors have read and agreed to the published version of the manuscript.

**Funding:** This research work has been supported by GATE project, funded by the Horizon 2020 WIDESPREAD-2018–2020 TEAMING Phase 2 program under grant agreement no. 857155, by Operational Programme Science and Education for Smart Growth under Grant Agreement no. BG05M2OP001-1.003-0002-C01 and by the Bulgarian National Science fund under project no. KP-06-N32/5.

**Data Availability Statement:** Not applicable.

**Conflicts of Interest:** The authors declare no conflict of interest.

### Appendix A

*Appendix A.1. Completeness Evaluation*

**Table A1.** Completeness evaluation of the original model. Results from assessment of the correspondence between the user requirements and the UML diagram of the original model. Type 1 error—items represented in the model but not corresponding to the requirements; Type 2 error– items represented the requirements but not in the model; Type 3 error—items that are represented in the model and correspond to the requirements but are defined inaccurately in the model.

| | Requirement | Is Satisfied | Error Type | Type 1 Errors |
|---|---|---|---|---|
| 1. | The model should have a temporal/historical dimension (analyzing patients' data over time). | Yes | - | Financial state listed in anamnestic data |
| 2. | The model should describe a patient profile containing medical data traditionally collected when neurodegenerative or cognitive disease is suspected. | Yes | - | Blood test contains clinician name |
| 3. | Each patient profile describes/belongs to a single patient. | Yes | - | |
| 4. | The model should present the functional specification for the patient record reflecting the described structure in horizontal and vertical plan as well as the correlations and interactions between the included properties | Yes | - | |
| 5. | The model should be oriented towards presenting data at the Patient level. | Yes | - | |
| 6. | The model aligned with the external knowledge sources, such as ADNI database. | Yes | - | |
| 7. | The model should set correlations that discover a relationship between the patient's status, historical data, and disease progression. | Yes | - | |
| 8. | The model should integrate information through biomedical abstractions, using proper medical terminology. | Yes | - | |

**Table A1.** *Cont.*

| | Requirement | Is Satisfied | Error Type | Type 1 Errors |
|---|---|---|---|---|
| 9. | The model should allow application for modeling all kind of cognitive disorders in unified way. | Yes | - | |
| 10. | The model should integrate diverse medical data at different levels of granularity. | Yes | - | |
| 11. | Each patient profile should have a unique identifier. | No | Type 2 | |
| 12. | Each patient profile should contain patient's personal data namely ethnicity, race, age, gender, name and main language. | Yes | - | |
| 13. | All patients should be anonymous. Therefore, their names should be anonymized or listed as initials. | Yes | - | |
| 14. | Each patient profile should contain anamnestic profile including medical history, lifestyle information, relevant socio-economic information and family history. | Partially | Type 3 | |
| 15. | Each patient profile should contain only one anamnestic profile that can be updated to reflect current state. | No | Type 2 | |
| 16. | Each patient profile should contain a record of previous treatments as well as their current treatment. | No | Type 2 | |
| 17. | Each medication should have a dosage in milligrams. | Yes | - | |
| 18. | Each treatment should have a date of prescription and duration. | Yes | - | |
| 19. | Each treatment is made of list of medications that were prescribed at some point of time. | No | Type 2 | |
| 20. | Each patient profile should contain a record of previous diagnoses as well as their current diagnose. | No | Type 2 | |
| 21. | Each patient profile can contain data from one or more check-ups that the patient attended. | Yes | - | |
| 22. | Each patient profile should contain results from clinical assessments the patient underwent. | Yes | - | |
| 23. | Each patient profile should contain information about existing comorbidities. | Yes | - | |
| 24. | Family history should include information on presence of neurodegenerative diseases in the family. | Partially | Type 3 | |
| 25. | Each anamnestic profile should have information about the length of their education listed in years. | Partially | Type 3 | |

**Table A1.** *Cont.*

| | Requirement | Is Satisfied | Error Type | Type 1 Errors |
|---|---|---|---|---|
| 26. | Each anamnestic profile should contain patient's handedness with possible values left dominant, right dominant or ambidextrous. | Yes | - | |
| 27. | Each anamnestic profile should contain patient's computer literacy with possible values none, low, average, expert. | Yes | - | |
| 28. | Each anamnestic profile should contain patient's employment status with possible values full time, part time, unemployed, or retired. | Yes | - | |
| 29. | Each anamnestic profile should contain patient's marital status with possible values single, married, divorced, widowed, in relationship. | Yes | - | |
| 30. | Each anamnestic profile should contain patient's living situation with possible values alone, with partner, with family, with caretaker. | Yes | - | |
| 31. | Each anamnestic profile should contain patient's place of residence with possible values city, near a city, remote area. | Yes | - | |
| 32. | Each anamnestic profile should contain patient's diet and diet specifics as a free text. | Partially | Type 3 | |
| 33. | Each anamnestic profile should contain patient's physical activity in terms of time with possible values less than one hour a week, between 2 and 5 h a week, more than 5 h a week. | Partially | Type 3 | |
| 34. | Each anamnestic profile should contain patient's alcohol consumption in terms of frequency with possible values once or twice a month, once a week, more than once a week. | Partially | Type 3 | |
| 35. | Each anamnestic profile should contain patient's current smoking status with possible values never smoked, smoking, quitted smoking. | Partially | Type 3 | |
| 36. | Each anamnestic profile should contain patient's current drug abuse status with possible values never used, on regular basis, recreational use, or quitted using drugs. | Partially | Type 3 | |
| 37. | Each anamnestic profile should contain patient's current sleep duration status with possible values about 8 h, less than 4 h, more than 9 h. | No | Type 2 | |
| 38. | Each anamnestic profile should contain patient's current sleep quality with possible values well-rested most of the days, feeling tired most of the days, experiences insomnia or sleep disruptions. | Partially | Type 3 | |

**Table A1.** *Cont.*

| | Requirement | Is Satisfied | Error Type | Type 1 Errors |
|---|---|---|---|---|
| 39. | Medical history should contain information about previous head traumas. | Yes | - | |
| 40. | Clinical assessments should include imaging assessments, neuropsychological assessments, CSF biomarkers assessments, blood tests and genetic tests. | Yes | - | |
| 41. | Possible comorbidities should include high blood pressure, dyslipidemia, diabetes, obesity, atrial fibrillation, other heart diseases, carotid stenosis, autoimmune disease or vasculitis, psychiatric disease, other significant diseases. | Yes | - | |
| 42. | Neuropsychological assessments should include cognitive assessments, neuropsychiatric assessments and assessments of daily living activities. | Yes | - | |
| 43. | Cognitive assessments should include Mini-Mental State Examination (MMSE), Montreal Cognitive Assessment (MoCA), Isaac's Set Test (IST), California Verbal Learning Test (CVLT), Brief Visuospatial Memory Test—Revised (BVMT-R), Symbol Digit Modalities Test (SDMT). | Yes | - | |
| 44. | Neuropsychiatric assessments should include Beck's Depression Inventory (BDI), Hamilton Anxiety Rating Scale (HAM-A), Fatigue scale (FSS), Neuropsychiatric Inventory Questionnaire (NPI). | Yes | - | |
| 45. | Daily living activities are assessed with Lawton Instrumental Activities of Daily Living (IADL). | Yes | - | |
| 46. | A result from MRI or CT imagining assessment should include not only the image itself but also a reading of the following characteristics: global cerebral atrophy, medial cerebral atrophy, posterior cerebral atrophy, assessment of white matter lesions as per Fazekas scale. | Yes | - | |
| 47. | Global cerebral atrophy should be rated on a scale between 0 and 3 (GCA scale) | Partially | Type 3 | |
| 48. | Changes in the white matter should be rated on scale between 0 and 3 for each component of the Fazekas scale—PVWM and DWM. | Partially | Type 3 | |
| 49. | Medial temporal atrophy should be rated on scale between 0 and 4 for each lobe (MTA score/Scheltens' scale). The average score of both lobes should be calculated, too. | Partially | Type 3 | |
| 50. | The average MTA score of both lobes should be also calculated. | No | Type 2 | |

**Table A1.** *Cont.*

| | Requirement | Is Satisfied | Error Type | Type 1 Errors |
|---|---|---|---|---|
| 51. | Posterior atrophy should be rated on a scale between 0 and 3 (Koedam score). | Partially | Type 3 | |
| 52. | CSF biomarkers should include total tau (t-tau), phosphorylated tau (p-tau), Aβ40, Aβ42, p-tau/t-tau ratio, Aβ42/Aβ40 ratio. | Yes | - | |
| 53. | Each CSF measurement should contain the name of the measured substance, unit of measurement, laboratory name and normative values. | Yes | - | |
| 54. | Each blood measurement should contain the name of the measured substance, unit of measurement, laboratory name and normative values. | Yes | - | |
| 55. | The chosen format of presentation of the model must be known/popular and understandable to the medical expert and technical staff to be successfully validated. | Yes | - | |
| 56. | The model must provide a complete and non-contradictory structure to be used in the selection of the proper database and its creation. | Yes | - | |
| 57. | The model should have a temporal/historical dimension (analyzing patients' data over time). | Yes | - | |

*Appendix A.2. Questionnaires Used for the Evaluation of 'Understanding'*

Appendix A.2.1. Participant Profile and Background

**Table A2.** Participant information questionnaire.

| Question | Answer Options |
|---|---|
| **What is the highest degree that you have completed?** | - Highschool<br>- Bachelor<br>- Master<br>- PhD |
| **What is the field of your studies?** | Free text |
| **How familiar are you with UML diagramming?** | - Not familiar at all<br>- I have heard about UML before<br>- I have some idea since I have used/created UML diagrams previously<br>- I am confident in my knowledge of UML diagramming |
| **How familiar are you with common medical concepts such as anamnesis, brain imaging, comorbidity, medical history, etc.?** | 1 (not at all)—5 (very familiar) |
| **How familiar are you with cognitive diseas-es?** | 1 (not at all)—5 (very familiar) |

Appendix A.2.2. Cloze Test

The Cloze test used in the present study featured 14 missing words that had to be completed. In some cases, there were several possible answers. In such cases all of the words are listed, separated with a slash. The full text with the completed answers is available below. Blanks are indicated by underlines.

The CogniTwin data model represents a digital patient profile that corresponds to a single patient. It is designed to hold data relevant to the diagnosis and observation of cognitive diseases, including Alzheimer's disease and dementia. Each patient profile can contain information from one or several visits/examinations. Therefore, the data model allows the storage of longitudinal data. Each patient profile contains personal information, such as patient initials, age and identifier/gender/ethnicity/race/language. The main components of the patient profile are—anamnestic profile, treatment plan and clinical profile. A single instance of ClinicalAssessmentsRecord is instantiated by ClinicalProfile entity under the property name recordOfClinicalAssessments and contains important data such as results from blood tests, csf tests and imaging assessments.

During an imaging assessment, the patient undergoes a scanning procedure that produces an image of his/her brain. This image is then interpreted by a trained physician. There are different types of imaging methods and sequences but the data model allows the storage of four types of images, namely—MRI T1, MRI T2, MRI Flair and CT*. The image and the interpretation are stored in an ImagingResult entity. Each patient has an imaging record which can contain any number of such records.

To interpret an image, a physician must examine the state of various brain structures for abnormalities such as lesions and deformities. Such evaluation is systemized with scales such as the Medial Temporal Atrophy Scale (MTA) or Fazekas scores. In the data model, most of the attributes representing results from such scales are of type numeric**. However, Fazekas scores make an exception since their possible values are encoded with enumeration.

\* accepted in any order

\*\* float and integer were also accepted as answers

### Appendix A.2.3. Problem-Solving Task

The problem-solving task included six questions, two of which (Q5 and Q6) required manipulation of the UML model and thus are referred as constructional. All questions and their respective base answers are shown in Table A3. Note that the grading of the answers did not look for sentence matching with the base answer rather it looked for match in the meaning.

**Table A3.** Questions included in the problem-solving task. The grading of each answer was based on the formulation of sample answers.

| Question | Base Answer |
|---|---|
| **Q1:** A healthy elderly patient goes for a regular check but expresses concerns about his/her mental health and memory. Discussing their current condition requires information about their physical and psychological state over the last year. Provided that the physician is using a system that implements the data model, is it possible to retrieve results from neuropsychological assessments that were conducted over the last year? Why? | Yes, it is possible. The data model supports storage of longitudinal data. |
| **Q2:** John wants to investigate whether depression can predict future onset of dementia. John has access to a system that is implementing the data model and has thousands of records. Is such a system useful to John in terms of finding appropriate data? Motivate your answer. | Yes, it it. The data model contains longitudinal data about the neuropsychological state of a patient as well as a record of diagnoses. |
| **Q3:** George is a researcher who is investigating the metabolic brain changes in dementia patients compared to cognitively healthy adults. For this purpose, John needs readings from positron emission tomography images (PET imaging) of brain metabolism. Can such information be stored in a system implementing the data model? Motivate your answer. | No. The provided data model allows the storage of four types of images—MRI T1, MRI T2, MRI Flair, CT. |

**Table A3.** *Cont.*

| Question | Base Answer |
| --- | --- |
| **Q4:** John is interested in the interaction between genetic predisposition and lifestyle factors in determining the risk of developing Alzheimer's disease. In particular, he wants to investigate whether individuals who have healthy lifestyle habits, family history of Alzheimer's disease and confirmed genetic predisposition are at same risk as equally genetically burdened individuals with unhealthy lifestyle. Can John find all of the data he needs in a system implementing the CogniTwin data model? | Yes. The data model stores lifestyle data (diet, sport, etc.) as well as data about the predisposition to Alzheimer's disease in terms of family history and confirmed by laboratory test genetic predisposition. |
| **Q5:** We want to extend the model to include results from measuring glucose levels in whole blood samples. Which entities do we need to change and how? | 1. Add glucose attribute to BloodStatus;<br>2. Add glucose to TestParameter Type. |
| **Q6:** How would we change the model if we want to be able to store results from measuring glucose levels, protein levels and red blood cell count in urine samples? | 1. Create new enityty UrineStatus with attributes glucose, protein and red blood cell count;<br>2. Add urine to SampleMaterial Type;<br>3. Add glucose, protein and red blood cell count to TestParameter Type. |

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
