# Peer review of "Cognitive Impairment and Dementia Data Model: Quality Evaluation and Improvements"

_computers, doi:10.3390/computers12020029_

Round 1

Reviewer 1 Report

The authors evaluate and improve a data model for cognitive impairment that they have previously presented. The article is sound, the methodology is clearly presented and the results support the conclusion. 

There are some unclear aspects that need to be cleared:

-for completeness criteria, what are the user requirements (they are described in the other article but this article should be self-contained) and how were they compiled?

-for the understandability criteria more information should be supplied regarding the participants sampling

My main concern with this article is that the authors already published another article with part of this work, the model itself, which makes the contribution for this journal article quite thin. It would have made the article more complete if part of the future work was actually done.

Reviewer 2 Report

This paper's topic is interesting. The quality of data models is an important and exciting topic. The author mixes data models and their evaluation. Evaluation of data models by survey makes no sense. The very base and essential issue is terminology. Authors understand UML as a data model, but it is not valid.  UML is a visual modelling language which contains 14 types (+/-) of models. This makes the paper unsuitable for review on Computers. The second aspect is a survey. For data model evaluation, it is not the best option. Some rigorous approaches will be better and replicable. 

Reviewer 3 Report

This paper seeks to discuss a common data model for a unified cognitive disease description. Although this study is significant, it is not addressed in considerable detail, particularly with regard to its methodology and method selection.

1. Specific names should not be mentioned in the choosing of a methodology when there are other , more reliable methodologies available and no justifications are provided.

2. In the introduction, the rationale for the problem is too general, and the significance and application of the research's scope are unclear. It should be more specific with scenarios and convincing argumentation, such as the author's prior study findings.

3. Very limited literature review; it appears that the use of the methodology proposed by D. Moody and G. Shanks must be contested by other methodologies. Additionally, it is required to elaborate on the data model quality evaluation.

4. It is important to examine the reliability of the methodology that has been selected in the analysis section.

Good luck!
